# The Impact of COVID-19 on New Kidney Cancer Diagnosis: Stage and Treatment in Northern Italy

**DOI:** 10.3390/ijerph20064755

**Published:** 2023-03-08

**Authors:** Lucia Mangone, Francesco Marinelli, Giulia Bonfante, Isabella Bisceglia, Fortunato Morabito, Cristina Masini, Franco Antonio Mario Bergamaschi, Carmine Pinto

**Affiliations:** 1Epidemiology Unit, Azienda USL-IRCCS di Reggio Emilia, 42123 Reggio Emilia, Italy; 2Unit of Urology, Azienda USL-IRCCS di Reggio Emilia, 42123 Reggio Emilia, Italy; 3Biotechnology Research Unit, Aprigliano (CS), 87051 Aprigliano, Italy; 4Medical Oncology Unit, Azienda USL-IRCCS di Reggio Emilia, 42123 Reggio Emilia, Italy

**Keywords:** kidney cancer, stage, treatment, COVID-19

## Abstract

This study aims to evaluate the impact of COVID-19 on new renal carcinoma (RC) diagnoses using data from the Reggio Emilia Cancer Registry in 2018–2020. A total of 293 RCs were registered, with roughly 100 cases yearly. The distribution by age shows a significant decrease in the 30–59 age group (33.7% in 2018, 24.8% in 2019, and 19.8% in 2020). The incidence of Stage I was 59.4%, 46.5%, and 58.2% in 2018, 2019, and 2020, respectively, whereas the Stage II rate had values of 6.9%, 7.9%, and 2.2% in the years 2018, 2019, and 2020, respectively. Slight non-significant variations were observed in Stages III and IV. Surgery was performed in 83.2% of cases in 2018, 78.2% in 2019, and 82.4% in 2020; the surgery distribution by stage showed no significant differences. Chemotherapy showed an increase in 2020, which was statistically significant only for Stage IV. The gender incidence trends over the last 25 years showed an increase in the male sex in the first period; then, a decline was documented, likely due to a decrease in cigarette consumption. In females, the trend was constant. The RC mortality trend significantly dropped in both genders over the entire study period.

## 1. Introduction

Renal carcinoma (RC) ranks ninth in frequency in Italy, with new cases exceeding 13,500 per year, primarily affecting males, with a gender incidence rate of 28.1 and 11.8 cases per 100,000 inhabitants in men and women, respectively [1]. Moreover, the incidence has shown an increasing trend yearly in men (+2.9%) and women (+2.2%). The same trend was also observed in other European countries [2,3], Canada [4], and the United States [5,6]. The incidence increases with age, reaching its peak in the eighth decade of life, and is significantly higher in men. RC is detected incidentally in over half of the cases and is confined to the kidney in 55% of cases, thus explaining the remarkable 5-year survival rate of 71% reached in Italy [1].

Environmental, behavioral, and genetic factors represent essential determinants in the pathogenesis of malignancy. Cigarette smoking is one of the main risk factors linked to nearly 40% of cases of RC in men [7]. Obesity represents an additional crucial risk element, primarily associated with RC devolvement in females [8]. Hypertension is related to 20–40% of RC [9], whereas the role of environmental exposure, especially in the workplace, remains somewhat controversial [10]. The increasing use of imaging (ultrasound, echotomography, magnetic resonance) has probably led to an increase in the incidental diagnoses of RC [11], especially concerning Stage I cases with small renal masses [12,13].

Mortality has decreased in many European countries [14], in the US [15], and in Canada [5]. Early diagnosis and therapy improvements largely contributed, especially in treating metastatic disease. The availability of targeted therapies, i.e., checkpoint inhibitors and immune-based combinations, have recently emerged, significantly improving the RC overall survival rates [16,17,18,19]. Stage, grading, and histotype remain the most important prognostic factors. Surgery represents the standard treatment of localized RC [1].

The impact of COVID-19 on new cancer diagnoses has been widely described in the literature. Liang et al. [20] highlighted the negative implications of SARS-CoV-2 infection on cancer patients in China. Specifically, the authors emphasized that neoplastic patients are more likely to be hospitalized and die in intensive care, especially for newly diagnosed patients. Further studies on the same topic subsequently demonstrated that COVID-19 delayed the diagnosis and treatment of the tumors, ultimately affecting the incidence and mortality rates [21,22,23].

During the pandemic, an increase in mortality among cancer patients has been detected, especially among old males, people with comorbidities and low-performance status, and smokers [24]. Furthermore, some Italian studies have shown an increased risk of hospitalization and death from COVID-19 in cancer patients compared to the general population. In particular, this was highlighted for lung cancer, breast cancer, and onco-hematological diseases [25].

The pandemic management has required the reallocation of already-insufficient resources. In addition, to make the already-precarious setting worse, services and departments other than the forefront of emergency management were slowed down or even closed. As a result, delays and bottlenecks accumulated along diagnostic/therapeutic procedures, and access was significantly cut-rated to essential visits. Accordingly, specialist and instrumental evaluations were reduced, leading to all of the above translating into a diagnosis reduction and delay. Similarly, a delay in surgical treatment accounted for the reduced availability of operating rooms, leading to a potentially worse patient clinical, functional, and emotional outcome. The consequent accumulation would weigh on the healthcare system in the future.

This work aims to evaluate the impact of the SARS-CoV-2 pandemic on newly diagnosed kidney cancers and their treatment in the province of Reggio Emilia.

## 2. Materials and Methods

This population-based cohort study used data from the Reggio Emilia Cancer Registry (RE-CR) approved by the provincial Ethics Committee of Reggio Emilia (Protocol no. 2014/0019740 of 4 August 2014). The RE-CR covers a population of 532,000 inhabitants. RE-CR is a high-quality CR since it updates the incidence data (extended to 2020), and more importantly, it is supported by a high percentage of microscopic confirmation (89% for kidney cancer) and a DCO (Death Certificate Only) rate below 0.1% [26]. The leading information sources of the RE-CR are anatomic pathology reports, hospital discharge records, and mortality data, integrated with laboratory tests, diagnostic reports, and information from general practitioners.

Kidney cancer cases were defined based on the International Classification of Diseases for Oncology, Third Edition (ICD-O-3) [27], as topography C64.9. The study encompassed all kidney cancer cases diagnosed in the 2018–2020 period with detailed reports on the stage (TNM 8th edition) [28], surgery, and therapy, including tyrosine kinase inhibitors with an anti-angiogenic mechanism (VEGFR-TKI) and immunotherapeutic drugs. All the information was collected by consulting the hospital’s medical records. Unfortunately, we have failed to collect radiotherapy information yet. To better understand the evolution of RC over the years, without focusing only on the last three years, we have extended the survey to a broader period. For this reason, the incidence and mortality trends over the previous 25 years have been analyzed by gender.

Descriptive analyses of RC patient characteristics were presented by year (i.e., 2018, 2019, and 2020). The proportions by the period of cancer diagnosis were calculated by sex, age, stage, method of diagnosis, surgery, and chemotherapy. We performed the nonparametric test for trends to determine the difference in these covariates between years.

The standardized incidence and incidence-based mortality rates of the last twenty-five years (1996–2020) were calculated by gender, adjusted to the 2013 European standard population, and calculated per 100,000 person-years. The population-based estimation rates derived from the province of Reggio Emilia data recorded on 1 January of each year.

Analyses were performed using STATA 16.1 software. In this study, we reported 95% confidence intervals (CI) and defined a *p*-value < 0.05 as statistically significant. Trends over time were analyzed by calculating the annual percent change (APC) in age-standardized rates using Joinpoint Regression.

## 3. Results

In 2018–2020, 293 RC cases were registered (101 in 2018, 101 in 2019, and 91 in 2020, respectively) (Table 1). The distribution by age shows a significant decrease in young people between 30 and 59 years. The incidence ranged from 33.7% in 2018 to 24.8% in 2019 and 19.8% in 2020. A slight increase was observed in the 60–79 age group from about 48% to 57% in the 80+ range, both of which are not significant. The distribution by sex shows a slight increase in 2019 and 2020 compared to 2018. The distribution by age fluctuates, moving from 59.4% in 2018 to 46.5% in 2019 and 58.2% in 2020 for patients in Stage I. Stage II shows even values of 6.9%, 7.9%, and 2.2%, respectively, in 2018, 2019, and 2020. Slight variations are also observed for Stages III and IV but are not significant. The diagnostic modality generally shows a slight decrease in histological confirmations in 2019–2020 compared to 2018. Surgery was performed in 83.2% of cases in 2018, 78.2% in 2019, and 82.4% in 2020. Overall, in 2019-2020, chemotherapy decreased (about 15%) compared to 2018 (20%).

Looking at the distribution of surgery by stage (Table 2), no significant differences are observed for Stage I (about 98% of cases undergoing surgery), Stage II (the numbers are minimal), and Stages III and IV. Therapy showed no changes in the first three stages, whereas Stage IV displayed a decrease in 2019 (65.2%) compared to 2018 (72.2%), and then increased in 2020 (80%); the change was statistically significant. Concerning the years reviewed in our work, 2018–2020, two antiangiogenic drugs, i.e., Sunitinib and Pazopanib, were chosen as the first-line therapies for fit and fragile patients, respectively. An immunotherapy approach with Cabozantinib was subsequently utilized for progressed cases. The first-line strategy needs to be updated since combining an antiangiogenic drug with immunotherapy entered the real-world algorithm.

The trend of incidence and mortality of RC in 25 years of incidence (Figure 1) shows, in males, an increase in incidence (APC 2.3%, 95% CI 0.6; 4.1), followed by a decrease in subsequent years, even if not significant (APC −3%, 95% CI −6; 0.2); in females instead, the trend remained constant in the 25 years examined (APC 0.1%, 95% CI −7; 1). A significant decline in mortality was observed in males over the whole period (Figure 1). Specifically, death incidence reduced from 12 per 100,000 in 1996 to 1 per 100,000 in 2020 (APC −3.3%, 95% CI −5.1; −1.5. The same phenomenon was observed in females (APC−4.5, 95% CI −6.2; −2.8).

## 4. Discussion

Our study aimed to evaluate the real impact of the COVID-19 pandemic on RC incidence in terms of disease stage and describe the implications for surgery and therapy.

The rapid spread of COVID-19 has led to a reconfiguration of the departments’ structure as well as healthcare personnel to cope with the crisis. Consequently, the clinical activity of disciplines not primarily involved in managing the pandemic has suddenly stopped. Just as all non-emergency procedures were postponed, urological activities, including elective surgeries, also fell sharply, raising concern about the risk of adverse events related to diagnostic and treatment delays [29,30].

During the pandemic, several national and international urological societies have published a series of recommendations to define the priorities of the different clinical and surgical activities during the COVID-19 emergency, aiming to stem the negative impact of this crisis [31,32,33]. However, although studies with the best evidence have been analyzed, most recommendations are ultimately based on third-level evidence or expert opinions.

These recommendations may differ based on the specific geographical, socioeconomic, cultural, and health scenarios in which they were drafted. During the pandemic, the European Association of Urology (EAU) recommends performing only emergency surgeries, which cannot be postponed for more than 24 h. Similarly, a high priority is deserved for those cases for which a delay of more than six weeks would most likely cause clinical damage (i.e., progression, metastasis, loss of organ function, or death). Cases with intermediate or low priority should be timely reallocated according to locally available resources [33]. Surgery could be unsafe in asymptomatic COVID-19-positive patients [34] since their rate of developing respiratory complications exceeds 50% peri-operatively, with a 1-month death rate of 38% [20]. The adverse outcomes mentioned above are mainly accounted for in elderly male patients or those with cancer or comorbidities.

Approximately 20% of all new cancer diagnoses yearly are in the genitourinary tract, accounting for more than a third of all new cancer diagnoses for men [35]. For uro-oncological patients, the most significant benefit in terms of survival derives from radical surgery, hence the concerns related to delays in treatment. Based on the knowledge gained on the natural history of urological cancers, evidence has been produced to support the idea that most uro-oncological surgeries can be postponed with some degree of safety to a historical period in which healthcare resources are limited [36]. Despite the recommendations of the Urological Societies, it is evident that the reduced availability of operating theaters has inevitably led to a delay in treating oncological diseases. In the Italian study published by Gontero et al., an alarming fact has emerged: in Piedmont and Valle D’Aosta, more than 50% of uro-oncological operations have suffered a delay of more than 30 days between diagnosis and treatment, which represents a reasonable period to treat all new diagnoses [29].

An Italian study shows that the cumulative delay between surgery and missed visits can lead to a knock-on effect on future patients, further exacerbating potential adverse events [29]. Since this is an unprecedented emergency and the duration is unknown, this delay could have worse repercussions in the long term than COVID-19 itself [37]. The most significant concern is the risk of progression and the substantial accumulation in sick patients awaiting surgical treatment, given the high incidence of these neoplasms [31]. In a recent review, Wallis et al. [38] highlighted how the risk of progression is related to the type and degree of the disease.

An English study [39] showed that for aggressive tumors, even a modest delay in surgical treatment leads to an increased death rate; for example, in patients with bladder, lung, ovarian, pancreatic, or stomach cancer in Stages II or III, six-month survival decreases by more than 30%. In comparison, three-month survival reduces by more than 17%. However, a six-month delay will significantly increase mortality even for common tumors with a relatively favorable prognosis.

An Italian study reports that during the lockdown (March–May 2020), there was a 45% drop in new diagnoses for cancer compared to the two years 2018–2019 [40]. In detail, the missed diagnoses concern skin cancers and melanomas (−57%), colorectal cancers (−47%), prostate cancers (−45%), and bladder cancers (−43.6%). Therefore, in Italy, 35.6% of missed cancer diagnoses are borne by the prostate and bladder; this figure highlights how uro-oncological diseases have been critically affected by the lockdown.

Our study showed a slight decrease in RC diagnoses in 2020 (91 cases) compared to 2018 and 2019 (101 patients). The reduction in incidence was nearly exclusively the prerogative of people younger than 60, for whom there was a higher chance of reaching the hospital during the pandemic than the elderly. Conversely, older patients continued to be admitted mainly for comorbidity treatment [41].

In 2020, no shift towards advanced-stage tumors was observed, a phenomenon also described for other tumor sites in Italy [42,43] and worldwide [44].

The diagnostic approach has not changed, as has surgery, whereas chemotherapy shows a decreasing trend that had already begun in 2019. Regarding treatments and staging, almost all Stage I RC cases underwent surgery without delay in 2020; chemotherapy experienced a slight increase, but only in phase IV.

The incidence trend shows a significant decrease in males since 2012: this reduction in incidence is primarily linked to reducing cigarette smoking, representing the leading RC risk factor. Smoking has decreased in our province for years, but only in males [45]. Mortality was stable over the entire period but fell in recent years (2010), a phenomenon related to the availability of new immunomodulating drugs. This has made it possible to improve the treatment even in advanced and locally advanced stages [46,47].

Among the study’s strengths, we would underline that this is a population-based rather than a hospital-based report. Moreover, conversely to typical CR studies in which data are usually collected only by site and morphology, CR variables were reached with the stage and treatment information herein. Furthermore, the 25-year follow-up gives us a broader temporal outlook of the phenomenon.

The assessment of the lockdown and vaccine’s effect on the later years of the COVID-19 pandemic could not be addressed due to the lack of more updated data. However, it should be emphasized that data from Cancer Registries usually have a 3–4-year latency between collection and publication. In Italy, the most recent data refer to 2020. Still, they are estimates referring to 2008–2016 diagnoses (see AIOM 2020, The Numbers of Cancer in Italy. https://www.aiom.it/wp-content/uploads/2020/10/2020_Numeri_Cancro-operatori_web.pdf (accessed on 24 January 2023)). Similarly, data are relatively updated (to 2019) internationally (see SEER, https://seer.cancer.gov/ (accessed on 24 January 2023)). The RE-CR made an additional effort to collect data for 2019 and 2020 to precisely reach the study’s endpoint aimed to analyze the effect of the pandemic on new cancer diagnoses.

A previous Italian work conducted on four centers using aggregated data reported a 40% discontinuity in treatments [48]. During the first wave of COVID-19 (March–May 2020) in our province, delays were quite limited. However, with the second wave of COVID-19 (September–December 2020), the increase in late-diagnosed tumors and the shift towards more complex laparoscopic surgery have led to a marked increase in the waiting list. Future studies can answer the question of potential adverse effects on patients’ outcomes.

Of note, the systemic oncological treatment start was timely managed in Stage IV patients. The delay in starting a systemic treatment was conversely due to the deferred diagnoses and the difficulty of carrying out tests during the first phase of the pandemic.

Among the study’s limitations, we recall a need for more information on surgical schedule delays and even on survival, considering the short follow-up time. Information on specific treatments, such as cryoablation, could help in evaluating the impact of COVID-19 on surgery, as highlighted in a recent Italian work [49]. However, we could not address the issue since no cryoablation treatments were carried out in our province. A similar surgical complication rate was accounted for in 2020 compared to previous years.

## 5. Conclusions

In conclusion, the study can help doctors shed light on the pandemic’s effect on rare cancer such as RC. The impact of COVID-19 has not had any adverse effects on diagnosis and treatment so far, but it cannot be excluded that any delays may occur in the coming years. The suspension of oncological screenings has caused a decline in new diagnoses of breast, colorectal, and cervical tumors; thus, the results of this study should not be generalized for other tumor sites. On the other hand, patients hospitalized for COVID-19 pneumonia could be incidentally more frequently diagnosed with kidney cancer.

Moreover, the result of this study could somehow be generalized to the rest of northern Italy, considering the similar incidence of kidney tumors, COVID-19 incidence and death rates, and healthcare organization. Conversely, our study analysis cannot be overturned in the southern areas since healthcare is not similarly and efficiently organized.

## Figures and Tables

**Figure 1 ijerph-20-04755-f001:**
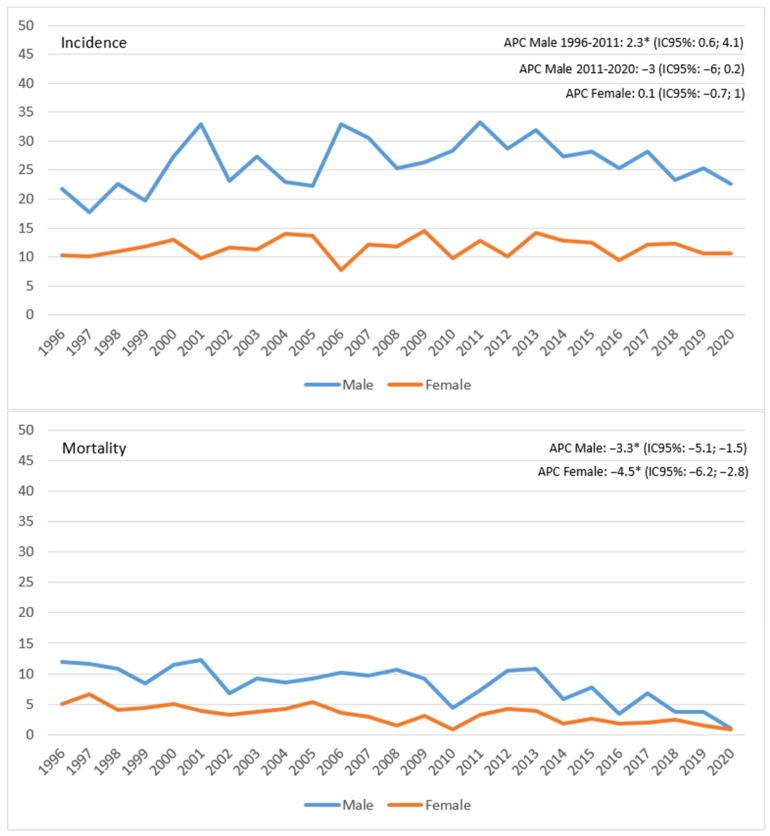
Reggio Emilia Cancer Registry. Years 1996–2020. Age-standardized incidence and mortality rates per 100,000 p-y in the province of Reggio Emilia. APC: annual percent change * APC is significantly different from zero at the α = 0.05 level.

**Table 1 ijerph-20-04755-t001:** Reggio Emilia Cancer Registry. Years 2018–2020. Number of cases by age, sex, stage, method of diagnosis, and therapy, per year.

	Years							Total (*n* = 293)	
	2018 (*n* = 101)		2019 (*n* = 101)		2020 (*n* = 91)				
	*n*	%	*n*	%	*n*	%	*p*-Value	*n*	%
Age at diagnosis									
30–59	34	33.7	25	24.8	18	19.8	<0.05	77	26.3
60–79	48	47.5	49	48.5	52	57.1	0.17	149	50.9
80+	19	18.8	27	26.7	21	23.1	0.44	67	22.9
Sex									
Males	62	61.4	67	66.3	60	65.9	0.48	189	64.5
Females	39	38.6	34	33.7	31	34.1	0.48	104	35.5
Stage									
I	60	59.4	47	46.5	53	58.2	0.82	160	54.6
II	7	6.9	8	7.9	2	2.2	0.15	17	5.8
III	13	12.9	16	15.8	16	17.6	0.34	45	15.4
IV	18	17.8	23	22.8	15	16.5	0.83	56	19.1
Unknown	3	3.0	7	6.9	5	5.5	0.39	15	5.1
Method of diagnosis									
Histological	92	91.1	88	87.1	79	86.8	0.35	259	88.4
Cytological	1	1.0	0	0.0	0	0.0	0.33	1	0.3
Clinical/instrumental	8	7.9	13	12.9	12	13.2	0.24	33	11.3
Surgery									
Yes	84	83.2	79	78.2	75	82.4	0.87	238	81.2
No	17	16.8	22	21.8	16	17.6	0.87	55	18.8
Chemotherapy									
Yes	21	20.8	16	15.8	13	14.3	0.21	50	17.1
No	79	78.2	76	75.2	76	83.5	0.37	231	78.8
Unknown	1	1.0	9	8.9	2	2.2	0.91	12	4.1

**Table 2 ijerph-20-04755-t002:** Reggio Emilia Cancer Registry. Years 2018–2020. Number of cases by stage and therapy per year.

	Surgery							Chemotherapy						
	2018		2019		2020			2018		2019		2020		
	*n*	% *	*n*	% *	*n*	% *	*p*-Value	*n*	% *	*n*	% *	*n*	% *	*p*-Value
Stage														
I	59	98.3	46	97.9	52	98.1	0.86	1	1.7	0	0.0	0	0.0	0.27
II	6	85.7	8	100	2	100	0.27	2	28.6	0	0.0	0	0.0	0.11
III	12	92.3	16	100	16	100	0.24	5	38.5	1	6.3	1	6.3	0.12
IV	6	33.3	8	34.8	5	33.3	0.94	13	72.2	15	65.2	12	80.0	<0.05
Unknown	1	33.3	1	14.3	0	0.0	0.40	0	0.0	0	0.0	0	0.0	/

* Percentages were calculated considering the total number of single-stage cancers per year.

## Data Availability

The data presented in this study are available on request from the corresponding author. The data are not publicly available due to ethical and privacy issues; requests for data must be approved by the Ethics Committee after the presentation of a study protocol.

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
