# Peer review of "The Impact of COVID-19 on New Kidney Cancer Diagnosis: Stage and Treatment in Northern Italy"

_ijerph, 2023, doi:10.3390/ijerph20064755_

Round 1

Reviewer 1 Report

Has Covid 19 had any impact on ablative treatments such as cryoablation? The Authors should discuss this topic (10.3390/medicina58081041). A revision of English language is required.

Author Response

Dearr Reviewer,

thanks for the comment, we hope that our answer is exhaustive.

Best regards,

Lucia Mangone

Comments and Suggestions for Authors

Has Covid 19 had any impact on ablative treatments such as cryoablation? The Authors should discuss this topic (10.3390/medicina58081041). A revision of English language is required.

RE: We Thank the Reviewer for the suggestion. Unfortunately, cryoablation treatments were not carried out in our province due to the lack of experience in this fiels. Indeed, patients were referred to selected centres with validated specific experience. In general, during 2020, to remedy the blockage of the operating rooms in our hospital, a surgical setting was rebuilt in another Covid-free hospital, continuing, albeit to a lesser extent, urological oncological surgery. 170 laparoscopic nephrectomies and 95 video-laparoscopic partial nephrectomies were performed in one year. Major complications did not present a higher incidence than in previous years.

Accordingly, we have added the following comments in the discussion section: ’……… Information on specific treatments, such as cryoablation, could help in evaluating the impact of Covid-19 on surgery, as highlighted in a recent Italian work [49]. However, we couldn’t address the issue since no cryoablation treatments were carried out in our Province. No additional surgery complications were accounted for in 2020 compared to previous years.’

Reviewer 2 Report

Please specify "chemotherapy"

Is it possible to evaluate the therapies by separating the patients who received immunotherapy from those who received target therapy?

Why did you not also consider the later years of the COVID-19 pandemic in order to assess the effect of lockdown and vaccines?

Please improve the discussion, for example PMID: 34707691

Author Response

Dear Reviewer,

thanks fort the comments. We hope that our answers are exhaustive.

Best regards,

Lucia Mangone

Comments and Suggestions for Authors

Please specify "chemotherapy"

RE: We thank the Reviewer for the request. We have changed the term of ‘chemotherapy’ in therapy, adding in the text specific information on the use of tyrosine kinase inhibitors or immunotherapy in these patients.

Is it possible to evaluate the therapies by separating the patients who received immunotherapy from those who received target therapy?

RE: We thank the Reviewer for the reasonable suggestion. Accordingly, the text has added the following comment: ‘……, Concerning the years reviewed in our work, 2018-2020, two antiangiogenic drugs, i.e., Sunitinib and Pazopanib, were chosen as the first-line therapies for fit and fragile patients, respectively. An immunotherapy approach with Cabozantinib was subsequently utilized for progressed cases. The first-line strategy needs to be updated since combining an antiangiogenic drug with immunotherapy entered the real-world algorithm….’

Why did you not also consider the later years of the COVID-19 pandemic in order to assess the effect of lockdown and vaccines?

RE: This is an exciting point addressed by the Reviewer. Unfortunately, we don’t have this important information. Hopefully, we will be able to answer the question after having available data by the end of 2023.

Nevertheless, we included in the Discussion section the following sentence: ‘…….The assessment of  the lockdown and vaccine's effect on the later years of the COVID-19 pandemic could not be addressed due to the lack of more updated data. However, the Reviewer's comment allowed us to highlight our work's strengths better. In this respect, the following sentences have been included in the Discussion section: ‘……..However, it should be emphasized that data from Cancer Registries usually have a 3-4 years latency between collection and publication. In Italy, the most recent data were published in 2020. Still, they are estimates referred to 2008-2016 diagnoses (see AIOM 2020, The Numbers of Cancer in Italy. https://www.aiom.it/wp-content/uploads/2020/10/2020_Numeri_Cancro-operatori_web.pdf). Similarly, data are relatively updated (to 2019) internationally (see SEER, https://seer.cancer.gov/). The RE-CR made an additional effort to collect data for 2019 and 2020 to precisely reach the study’s endpoint aimed to analyze the effect of the pandemic on new cancer diagnoses.

Furthermore, our study includes individual population data and not aggregated data, such as a previous Italian work conducted on 4 centers but on aggregated data which reported a 40% discontinuity in treatments [48]. During the first wave of Covid-19 (March-May 2020) in our Province, delays were quite limited. However, with the second wave of Covid 19 (September-December 2020), the increase in late-diagnosed tumors and the shift towards more complex laparoscopic surgery have led to a marked increase in the waiting list. Future studies can answer the question of potential adverse effects on patients’ outcomes.

Noteworthy, the systemic oncological treatment start was timely managed in stage IV patients. The delay in starting a systemic treatment was conversely due to the deferred diagnoses and the difficulty of carrying out tests during the first phase of the pandemic.

Please improve the discussion, for example PMID: 34707691

RE: According to the Rewiewer request, we added in the discussion section the following comments on the article the referee alluded to: ‘….  A previous Italian work conducted on 4 centers using aggregated data reported a 40% discontinuity in treatments [48]. During the first wave of Covid-19 (March-May 2020) in our Province, delays were quite limited. However, with the second wave of Covid 19 (September-December 2020), the increase in late-diagnosed tumors and the shift towards more complex laparoscopic surgery have led to a marked increase in the waiting list. Future studies can answer the question of potential adverse effects on patients’ outcomes.

Noteworthy, the systemic oncological treatment start was timely managed in stage IV patients. The delay in starting a systemic treatment was conversely due to the deferred diagnoses and the difficulty of carrying out tests during the first phase of the pandemic.

Reviewer 3 Report

Der Authors congratulation for this interesting  paper about a timeless topic : kidney cancer and covid 19.

I have no major query. 

Author Response

Dear Reviewer,

thank you very much for your appreciation of our work!!

Best regards,

Lucia Mangone

Comments and Suggestions for Authors

Der Authors congratulation for this interesting  paper about a timeless topic : kidney cancer and covid 19.I have no major query. 

RE: We thank the Rewiever for the positive comment.

Round 2

Reviewer 2 Report

None to update